# Effects of Proctoring on Online Intelligence Measurement: A Literature Overview and an Empirical Study

**DOI:** 10.3390/jintelligence13090110

**Published:** 2025-08-30

**Authors:** Vsevolod Scherrer, Nicolai Petry, Moritz Breit, Julian Urban, Julian Preuß, Franzis Preckel

**Affiliations:** Department of Psychology, Trier University, D-54286 Trier, Germany; s1nipetr@uni-trier.de (N.P.); breitm@uni-trier.de (M.B.); urbanj@uni-trier.de (J.U.); preuss@uni-trier.de (J.P.); preckel@uni-trier.de (F.P.)

**Keywords:** proctoring, intelligence, cheating, remote, literature review

## Abstract

Remote intelligence testing has multiple advantages, but cheating is possible without proper supervision. Proctoring aims to address this shortcoming, yet prior research on its effects has primarily investigated reasoning tasks, in which cheating is generally difficult. This study provides an overview of recent research on the effects of proctoring and on studies in intelligence test settings. Moreover, we conducted an empirical study testing the effects of webcam-based proctoring with a multidimensional intelligence test measuring reasoning, short-term memory, processing speed, and divergent thinking. The study was conducted in a low-stakes context, with participants receiving a fixed payment regardless of performance. Participants completed the test under proctored (*n* = 74, webcam consent), unproctored random (*n* = 75, webcam consent), or unproctored chosen (*n* = 77, no webcam consent) conditions. Scalar measurement invariance was observed for reasoning, processing speed, and divergent thinking, but not for memory. Proctoring had no significant main effect on test performance but showed a significant interaction with test type. Proctored participants outperformed the unproctored chosen group significantly in divergent thinking and scored descriptively higher in reasoning and processing speed, but slightly lower in memory. Observable cheating under proctored conditions was rare (4%), mostly involving note-taking or photographing the screen. We conclude that proctoring is crucial for easily cheatable tasks, such as memory tasks, but currently less critical for complex cognitive tasks.

## 1. Introduction

Intelligence testing is one of the cornerstones of psychological assessment. It is widely used to support placement and intervention decisions in a variety of applied settings ([30]; [34]; [46]). Furthermore, intelligence testing is employed in research spanning numerous academic disciplines, including psychology, education, sociology, economics, and medicine ([16]; [28]). Intelligence tests assess maximum performance, meaning they aim to capture the highest level of cognitive performance the test taker can achieve. Therefore, underperformance due to factors such as low motivation or adverse circumstances during testing, as well as artificial increases in performance due to drug use or cheating, pose a threat to their validity. In intelligence testing, cheating refers to intentional test-taker behavior that violates standardized conditions and undermines test fairness and validity. In online testing, this can include behaviors such as the use of unauthorized aids like internet searches, dictionaries, or calculators ([44]). Other forms include collaborating with others during a test, having another person complete the test, gaining prior access to test questions or materials, and using physical resources such as books, cameras, or handwritten notes during an online assessment ([35]).

To minimize adverse circumstances and cheating, intelligence tests are usually conducted in controlled environments under the supervision of a trained test administrator. However, the rise in computer-based assessment has made it possible to instead test individuals using their computers in their own homes (i.e., remote testing). This is an attractive option due to lower costs and reduced participation burden, especially in research contexts where large samples for cognitive testing are often needed, but difficult to obtain. At the same time, remote testing hinders environmental control and supervision, particularly raising concerns about cheating. Various methods have been developed to prevent or detect cheating, such as honor codes (e.g., [23]; [45]), specialized software environments (e.g., [29]), and statistical cheating detection (e.g., [24]). One of the most common methods is proctoring, where the behavior and environment of test subjects is observed directly during testing, including webcam recording of the test taker, screen capture, or keystroke logging ([44]). Implementing a proctoring solution in remote testing requires participants’ willingness to be monitored via a webcam feed and increases costs and participation burden compared to unproctored assessments, which partially negates the advantages of remote testing. Therefore, understanding the effects, effectiveness, and necessity of proctoring in remote testing is of interest to researchers and practitioners who must decide on the mode of testing for their study or diagnostic application.

Previous studies examining the effects of proctoring on intelligence testing have predominantly focused on reasoning tasks, which are not prone to cheating. However, a wide range of cognitive tasks exist that are relevant both for cognitive research and for comprehensive assessments of cognitive abilities in applied contexts. In this study we therefore investigated the effects of proctoring on four distinct abilities: reasoning, short-term memory, processing speed, and divergent thinking. Moreover, previous research usually compared test results between proctored and unproctored conditions. A precondition for doing so is that the test measures cognitive abilities equivalently across conditions (i.e., measurement invariance). There are very few findings for measurement invariance in proctoring studies. In our study, we therefore tested the measurement invariance of the intelligence test across different conditions to contribute to closing this gap.

### 1.1. Proctoring

Proctoring is the practice of supervising test takers during online exams or cognitive testing to prevent or detect dishonest behavior, i.e., cheating. One proctoring method is live video observation, in which a test administrator watches the test taker and their environment via a webcam feed. In some cases, the feed is recorded and watched later. A second method is artificial intelligence (AI) based proctoring, where the video feed is observed by a trained AI system that flags potential instances of cheating ([25]). In all cases, the aim is to deter cheating by making test takers aware that they are being observed and to detect any cheating.

#### 1.1.1. Reviews and Meta-Analyses on the Effects of Proctoring

[44] ([44]) conducted a meta-analysis of the impact of proctoring on ability assessments, including 49 studies published between 2001 and 2017. The analysis covered studies of low- and high-stakes testing situations involving paper-and-pencil or computerized tests. Most of the studies were conducted in educational or research contexts, while the remainder came from industrial and organizational contexts. The results showed that performance on unproctored assessments was superior to performance on proctored assessments, with a modest standardized mean difference effect size of *d* = 0.20. A key limitation, however, is that the original studies did not report measurement invariance of the tests across conditions. As a result, measurement invariance could not be investigated in the meta-analysis. Note that measurement invariance is not only a prerequisite for comparing means between groups ([7]; [6]) but also allows researchers to examine which properties of individual indicators differ between proctoring conditions. For example, if some test items are easier to cheat on due to higher searchability, these items would be expected to be flagged in measurement invariance testing by showing higher intercepts on the intelligence factor in the non-proctored condition. This would indicate that such items, relative to all other items, are easier to solve in the non-proctored condition than in the proctored condition. Therefore, investigating measurement invariance is essential when comparing proctored and non-proctored conditions.

Moderator analyses identified online searchability as the only significant moderator, revealing larger mean score differences for tasks that could be easily solved through internet searches. This finding led [44] ([44]) to recommend unproctored settings only for tasks that cannot easily be solved using online searches. No significant effects were found for the other moderators: stakes, countermeasures to cheating, and test mode (i.e., paper–pencil vs. computerized). The lack of significant differences in the results of high- and low-stakes conditions indicated that cheating may occur in both conditions when unproctored. In addition to cheating, the authors proposed reduced test anxiety in unproctored participants as an explanation for score differences between proctored and unproctored conditions.

Following the shift towards online educational settings prompted by the COVID-19 pandemic, [35] ([35]) published a systematic review comparing proctored and unproctored exams. The review incorporated 15 quantitative, qualitative, and meta-analytic articles published between 2004 and 2020. Most of these studies examined college students in the USA, except for [12] ([12]) and [44]’s ([44]) meta-analysis. Öncül found mixed results regarding differences in test scores between proctored and unproctored exams. Eight studies reported a significant difference, with unproctored exam scores being higher in seven of them, while the remaining studies found no difference. The potential reasons for these discrepancies in results were largely unexplored.

#### 1.1.2. Overview of Recent Studies

In light of rapid advancements in the field in recent years, we searched for empirical studies published since Steger et al.’s meta-analysis in 2018 [44] ([44]). Table 1 presents 15 studies that were identified through an online search. Only four of these studies were also included in [35] ([35]). Most of the recent studies focused on academic testing and were conducted in high-stakes academic settings, involving university students as participants. The majority examined samples from the US, with fewer studies from China, Indonesia, Mexico, Norway, and Spain. The studies varied in terms of the assessment location (i.e., on-site vs. remote) and the proctoring solutions used (i.e., webcam recording vs. automated proctoring software vs. on-site proctoring). In addition to reporting differences in test performance between proctored and unproctored conditions (14 studies), some studies also reported evidence of cheating (5 studies), differences in test duration (4 studies), and measurement invariance findings (1 study).

Of the 14 studies that examined differences in test performance, the majority (9; 64%) reported poorer test performance in proctored than unproctored conditions ([1]; [2]; [5]; [9]; [10]; [12]; [32]; [37]; [39]). This difference was sometimes more pronounced for more difficult questions ([1]; [5]) or when the test situation was perceived as high-stakes ([1]). Two studies found no difference in test performance between the two conditions ([4]; [31]), while the remaining three studies produced mixed results ([15]; [33]; [53]) with two reporting better performance in the proctored condition for some of their samples or tests ([15]; [53]).

Although better performance in unproctored tests was sometimes considered to be an indication of cheating, only five studies have explicitly investigated cheating behaviors, all in high-stakes settings ([4]; [5]; [10]; [31]; [54]). These studies generally reported low levels of cheating. However, in a longitudinal study, [5] ([5]) observed that the score advantage of the unproctored condition, the number of cheaters, and the benefit of cheating increased over the course of the semester, which the authors explained by an increase in cheating proficiency. Focusing exclusively on cheating, [54] ([54]) surprisingly reported a greater frequency of cheating in the proctored condition than in the unproctored condition. The authors recommended adding academic integrity reminders before the exam, which significantly reduced cheating in both the proctored and unproctored conditions.

In terms of test duration, participants were generally found to perform faster in proctored conditions than in unproctored ones ([1]; [9]; [53]). [9] ([9]) found that participants in the proctored conditions took only half as long to complete the exam. Conversely, [12] ([12]) reported no difference between the two settings but found that participants spent more time on more difficult tasks in the unproctored condition compared to the proctored condition.

Finally, only [31] ([31]) examined measurement invariance in their analyses, reporting strict measurement invariance over conditions. Of note, the lack of measurement invariance testing limits the interpretability of the reported differences in performance levels in most of the studies reported here. Seven years after [44] ([44]) raised the issue in their meta-analysis, this frequent omission remains a prevailing limitation of proctoring research.

#### 1.1.3. Proctoring in Intelligence Testing

Most studies included in the meta-analysis by [44] ([44]), the review by [35] ([35]), and our own review of recent studies focused on academic testing. The generalizability of these results to intelligence testing, particularly in low-stakes research settings, has rarely been examined, with a few notable exceptions. Table 2 presents nine studies that investigated the effects of proctoring in intelligence test settings. Most of them used reasoning tests, assessing figural content such as Raven’s matrices, verbal content such as word similarities, or numeric content such as number reasoning. However, other tests such as knowledge questions, mental rotation, and creativity tests were also applied. The investigated samples were diverse, as were the countries of sample origin (3 × USA, 2 × UK, Germany, Norway, Netherlands, Singapore). Five studies examined low-stakes testing situations, while four focused on high-stakes situations involving job applicants. Most studies compared on-site proctoring with online, unproctored testing conditions. In addition to reporting differences in average test performance between proctored and unproctored conditions (9 studies), some studies also reported evidence of cheating (5 studies), mean differences in test duration (1 study), and measurement invariance findings (2 studies).

Regarding average test performance, most studies reported no significant differences between proctored and unproctored conditions (5 studies: [20]; [26]; [31]; [41]; [47]). Other studies found mixed results, with unproctored participants performing better on certain tests or indicators but showing no differences or even worse performance on others (3 studies: [23]; [51]; [52]). [8] ([8]) reported better results in the proctored condition. [23] ([23]) considered the searchability of test content (e.g., knowledge questions versus reasoning tests) as a moderator and found that the advantage for the unproctored condition occurred only in the high searchability condition. This finding may partially explain why most studies found no difference regardless of test stakes, as most studies used reasoning tests with low online searchability.

Consistent with the lack of performance benefits in unproctored settings, most studies revealed limited or no evidence of cheating ([26]; [31]; [52]). [41] ([41]) observed moderate indication of cheating. [23] ([23]) reported higher levels of cheating in the unproctored condition than in the proctored condition using a highly searchable test.

Only one study ([52]) examined test duration and found mixed results. In one sample, unproctored participants had shorter test durations, whereas in another sample, they took longer to complete the test compared to proctored participants. Two studies ([31]; [52]) considered measurement invariance. [31] ([31]) observed the strict measurement invariance for their test. [52] ([52]) found that 8 out of 30 items showed differential item functioning. Notably, some items were slightly easier in the proctored condition, while others were easier in the unproctored group, which argues against systematic cheating across the entire scale.

#### 1.1.4. Summary and Open Questions

Overall, the available evidence suggests small yet significant performance advantages in unproctored conditions. However, this advantage only seems to exist for tests where answers can easily be found online, such as academic exams or knowledge tests. The advantage was not found for most intelligence tests, particularly reasoning tests, which were by far the most prevalent test format. Cheating levels were found to be low in all conditions. Test duration and measurement invariance were rarely considered in the studies.

Although the available findings provide valuable guidance for researchers and practitioners, some questions remain. First, previous research has only distinguished between searchable and non-searchable test formats. However, there are other ways to cheat besides looking up answers to knowledge questions online. For instance, memory tests allow cheating through note-taking or taking photos of the screen. Other tasks, such as verbal reasoning or divergent thinking tasks, can be solved with the help of large language models (LLMs), like ChatGPT or Claude. Finally, some tasks, such as for example, processing speed tasks, which are simple and scale through response speed, or figural matrix tasks, which require complex inductive reasoning that cannot be easily conveyed, do not lend themselves to cheating because any attempt incurs time costs without improving performance. Thus, when it comes to distinguishing between tasks, the concept of searchability falls short. A more general classification based on cheatability can cover additional cases.

Second, most findings on average performance differences between proctored and unproctored conditions do not rely on prior tests of measurement invariance. This is problematic because violations of scalar measurement invariance can bias mean-level comparisons ([3]; [42]). Measurement invariance examines whether the same set of indicators measures an identical latent construct equivalently across different groups or conditions. Different levels of measurement invariance can either be established or violated ([7]; [27]). The lowest level, configural invariance, implies that the same measurement model fits both conditions. Metric invariance assumes equal factor loadings across conditions, allowing comparisons of latent correlations and regressions between groups in different conditions. Scalar invariance tests whether intercepts of indicators are identical across conditions, allowing comparisons of latent means. Non-invariant intercepts across conditions suggest that specific indicators differ in difficulty relative to the overall scale. For example, in unproctored conditions, certain indicators might become easier due to increased opportunities for cheating (e.g., higher online searchability compared to other items). If scalar measurement invariance is not established, observed mean differences between groups remain ambiguous, as it is unclear whether differences reflect true differences in the latent construct or differences arising from indicators measuring distinct constructs across conditions. Therefore, previous findings regarding performance differences between proctored and unproctored conditions should be interpreted cautiously, given the possibility that identical performance indicators might not measure the same construct in both conditions ([44]).

### 1.2. The Present Study

Previous studies examining the effects of proctoring on intelligence testing have predominantly focused on reasoning tasks. However, a wide range of other cognitive tasks exist that are relevant both for cognitive research and for comprehensive assessments of cognitive abilities in applied contexts. Cognitive tasks such as memory or processing speed tasks differ in the ease with which participants can engage in cheating behavior during remote testing and in the types of cheating strategies that may be employed to solve them (e.g., note-taking vs. online search). To consider differences in cheatability, we investigated the effects of proctoring on four distinct intelligence abilities (i.e., reasoning, short-term memory, processing speed, divergent thinking). For reasoning and divergent thinking, we expected low cheatability, as cheating in these tasks would likely require support from AI tools or assistance from other individuals. For short-term memory, we expected high cheatability due to the potential for note-taking, either by writing down or photographing items with a mobile phone. For processing speed, we expected no cheatability, as there are no known cheating strategies that would meaningfully improve test performance.

We used a typical research scenario, involving online remote testing under low-stakes conditions. Specifically, we compared three experimental conditions: a proctored condition with webcam video recording, an unproctored random condition (UP random; participants were generally willing to be proctored with webcam video recording but were randomly assigned to not being proctored), and an unproctored chosen condition (UP chosen; participants explicitly declined proctoring with webcam video recording and were therefore not proctored). Including these three conditions was necessary for two reasons: First, it allowed us to experimentally assign participants to either a proctored or unproctored setting. Second, it enabled us to investigate whether individuals who explicitly chose not to be proctored systematically differed in performance from those who agreed to proctoring. In total, we addressed four research questions.

First, for each of the four ability test scales we examined three levels of measurement invariance (configural, metric, and scalar) across conditions. To the best of our knowledge, only two studies have investigated measurement invariance in intelligence testing across proctoring conditions before ([31]; [52]). Norrøne and Nordmo found support for the highest level of invariance, but this was examined only for assessments of reasoning ability, where cheatability is low. Wright et al. found that certain items of a speeded application test showed different psychometric properties across proctoring conditions. Therefore, we made no specific a priori assumptions regarding measurement invariance for the current study but assessed measurement invariance as an open research question.

Second, we investigated mean differences in test performance across conditions and whether the effects of proctoring varied by ability test (i.e., the interaction between condition and test type). For short-term memory tests, we hypothesized that mean scores would be lower in the proctored condition compared to both the UP random and UP chosen conditions, due to the possibility of cheating through note-taking in unproctored settings. For reasoning and divergent thinking tests, we made no specific a priori predictions, as cheating is possible but considered rather difficult. For processing speed, we expected no differences between proctoring conditions, as cheatability for this type of task should not be possible.

Third, we examined test duration across proctoring conditions as an open research question. Although cheating could theoretically impact test duration, existing empirical evidence regarding this relationship is inconclusive. It should be noted that only the total test duration was available in our study, so we were unable to distinguish test times for the individual ability tests.

Fourth, we explored how many participants demonstrated cheating behavior in the proctored condition with video monitoring. According to previous research (e.g., [26]; [31]; [52]), cheating is relatively rare. However, little is known about the prevalence of cheating in low-stakes remote intelligence testing with webcam proctoring.

## 2. Methods

### 2.1. Participants and Procedure

Data collection for the present study was conducted online between late April and early June 2025, using the Prolific platform as part of a larger data collection aimed at piloting the new digital intelligence test BIS+ (Berlin Structure-of-Intelligence Test Plus; developed by the authors of this paper). Prolific is an online platform through which participants can take part in scientific studies or commercial surveys for monetary compensation. All participants were recruited from Germany and self-reported fluency in German was a pre-requirement for study participation. The data collection was approved by the ethics committee of Trier University (protocol numbers 46/2025).

Overall, 266 test takers started the study; 40 of them dropped out and did not complete the tasks (*n* = 13 in the proctored condition; *n* = 9 in UP random condition; *n* = 18 in UP chosen condition). Thus, valid data from a total of 226 participants were available for analysis (age: *M* = 29.61, *SD* = 6.41, Min = 20, Max = 44; gender: 75 female, 147 male, 4 non-binary).

Participants were able to choose the day and time of their participation independently. The study description clearly communicated that participation required approximately 155 min of uninterrupted work in a quiet environment with full concentration. Participants received compensation of £25.84 (£10 per hour). Note that this fixed-amount payment structure, without additional incentives based on performance, may encourage participants to complete the study quickly and “well enough” to be approved, rather than to maximize accuracy. Participants were informed that attentive and earnest participation would help ensure the quality and validity of the test. Additionally, participants were offered individual feedback on their results upon request after completing the study. Participants were also informed that compensation would only be provided after the researchers verified serious participation. Individuals who terminated their participation prematurely did not receive compensation.

Participants who consented to webcam video recording on Prolific were randomly assigned to either (1) the proctored condition or (2) the UP random condition. In the proctored condition, participants were informed prior to testing that they would be recorded during the testing, were indeed recorded during the test, and the videos were subsequently reviewed by our research team. Additional proctoring methods such as screen capture or keystroke logging were not implemented due to limitations in technical feasibility within the Pavlovia system and their potential to reduce participant acceptance. In the UP random condition, participants were informed prior to testing that they would not be recorded, and no video recording took place. Finally, the third condition included participants who indicated on Prolific that they generally did not consent to video recording; these individuals were not recorded (UP chosen). The distribution across conditions was as follows: proctored condition (*n* = 74), UP random condition (*n* = 75), and UP chosen condition (*n* = 77).

Apart from recording or not recording the participants via webcam, the conditions did not differ. Participants first completed a short questionnaire assessing their self-estimated intelligence across various intelligence facets. Subsequently, participants completed the full BIS+ test, which was administered in two parts separated by a short break. Finally, participants completed a second questionnaire that assessed data on demographics, personality traits, and motivational variables.

### 2.2. Materials

#### 2.2.1. Intelligence Test

Intelligence was assessed using the BIS+, a digital, revised, and shortened adaptation of the BIS-HB ([22]) that is currently under development. It measures intelligence based on the Berlin Intelligence Structure Model (BIS; [21]; see Figure 1). The BIS conceptualizes intelligence along four operative abilities, reasoning (9 tasks in the present study), short-term memory (9 tasks), processing speed (9 tasks), and divergent thinking (6 tasks), fully crossed with three content domains: verbal (12 tasks), numeric (12 tasks), and figural (9 tasks). Thus, each process is assessed with material from every content domain, and each domain is assessed across all operative abilities, which are described in Table 3. In this study, we focus solely on the four operative abilities. To estimate reliability, we constructed parcels that each included one figural, one verbal, and one numerical task from each operative ability. The resulting Cronbach’s alpha/Omega values were α = 0.86/ω = 0.86 for reasoning, α = 0.85/ω = 0.85 for short-term memory, α = 0.93/ω = 0.93 for processing speed, and α = 0.71/ω = 0.71 for divergent thinking.

#### 2.2.2. Descriptive Information

Gender was assessed by asking participants to indicate their gender, with response options *Female*, *Male*, and *Non-Binary*. Current employment or student status was measured by asking participants to select the option that best described their situation: *School student, Vocational training, University student, Full-time employed, Part-time employed, Self-employed, Military/Civil/Social Service, Unemployed, Incapable of working, Homemaker,* or *Refugee.* Educational degree was assessed by asking participants to indicate the highest degree they had obtained. Response options were: *No school degree*, *Lower secondary school*, *Intermediate school degree*, *High school diploma (Abitur)*, *University degree*, *Doctorate (PhD)*, and *Other degree*. Parental education was measured using the same response categories as the participant’s own education. Primary language was assessed with an open-ended question (“Please indicate your primary language”), and answers were subsequently categorized for analysis. Finally, urbanity of residence was assessed by asking participants to indicate the type of area in which they live: *Village (<3000 inhabitants)*, *Small town (≥3000; <15,000 inhabitants)*, *Town (≥15,000; <100,000 inhabitants)*, *Large city (≥100,000; <1,000,000 inhabitants)*, and *Metropolis (≥1,000,000 inhabitants)*.

### 2.3. Data Analyses

Analysis code and dataset are available online on: https://osf.io/yznct/?view_only=66c66d3627104459926aaf4afa265c74 (Accessed on 17 August 2025). The data analyses were conducted with R 4.4.1. ([36]) using the following R packages: apaTables 2.0.8 ([43]), car 3.1.2 ([17]), dplyr 1.1.4 ([49]), flextable 0.9.6 ([14]), ggplot2 3.5.1 ([50]), lavaan 0.6.18 ([40]), psych 2.4.3 ([38]), purr 1.0.2 ([18]), readr 2.1.5 ([48]), and officer 0.6.6 ([13]).

#### 2.3.1. Preliminary Analyses

In preliminary analyses, we tested whether the proctoring conditions differed with respect to demographic variables. This step was necessary to ensure that any observed differences in intelligence tests were not simply due to systematic differences in the composition of the groups (e.g., notably younger participants or participants with substantially lower education in one condition). Age was analyzed using one-way ANOVAs with condition as the predictor. Categorical variables such as gender, current employment or student status, highest education, parental education, and primary language were tested using χ^2^ tests. For each variable, all categories that accounted for less than 5% of cases in the total sample (i.e., less than 12 participants) were combined into an “Other” category to ensure a robust estimation of the χ^2^-statistic. If the resulting “Other” category accounted for less than 5% of the total sample, it was excluded from the χ^2^-analyses. For example, in the case of gender, the non-binary category was excluded from the χ^2^-tests, as it represented only 1.77% of all cases (i.e., 4 out of 226 participants).

#### 2.3.2. Reviewing Proctoring Records

Similarly to [44]’s ([44]) recommendation to control post hoc for cheating behavior, we reviewed the available video recordings for evidence of participant cheating to ensure the validity of our data.

Participants were informed that webcam proctoring was applied to create a standardized testing situation and to ensure that the test was completed independently and without impermissible aids. The rules were stated in general terms rather than as an exhaustive list of specific permissible and impermissible behaviors. A standardized cheating protocol was developed that included clear indicators of cheating specific to our remote testing environment (e.g., discussing solutions with another person, another person providing an answer unsolicited, taking a photo of the screen with a cell phone, using a cell phone to solve a task, consulting a cheat sheet, writing notes, or self-disclosing one’s own cheating). Additionally, we recorded other relevant events that might impact the participant or the testing situation but could not be clearly classified as cheating (e.g., leaving the testing area, background noise, wearing headphones). All predefined options for cheating and other relevant events were available for selection in separate drop-down menus in an excel sheet. If none of the predefined options were applicable, or if additional information was necessary for evaluation, the observer could enter text in a designated field. For each event, the exact start time within the video was also documented.

The video recordings were reviewed by a trained student assistant and one co-author, each of whom was responsible for approximately half of the recordings. To prepare and obtain an initial overview, four video recordings were evaluated by both raters on a trial basis, followed by a detailed briefing and joint discussion. To determine the consistency between the raters, the video recordings of ten randomly selected participants were viewed by both raters. Cheating and non-cheating participants were identified with 100% consistency. The identification of other relevant events with major impact also showed perfect consistency across raters.

For each participant a separate digital cheating protocol was created. The video recordings, which typically consisted of two separate parts per participant (i.e., the first and second part of the intelligence test), were initially played at normal speed and then accelerated to eight times the speed within the first minute. At eightfold speed, audio output was not available, so only the visual stream could be used to identify relevant events. If irregular or notable events were detected (e.g., a participant appeared to be speaking), the video was paused, and the relevant segment was reviewed at normal speed. Sometimes (see below) the video stream, audio output, or segments of the video were missing. In such cases, the assessment was limited to the available material, and when only audio was present, the playback was slowed accordingly.

#### 2.3.3. Main Analyses

##### Measurement Invariance

We conducted a series of confirmatory factor analyses (CFAs) to evaluate configural, metric, and scalar levels of measurement invariance, using a stepwise approach and testing each intelligence ability separately.

Configural measurement invariance was tested by modeling each intelligence ability as a one-factor model, with each ability represented by its respective indicators (i.e., test tasks). Figure 2 illustrates, as an example, the modeling of the reasoning factor by its nine manifest indicators. As each indicator also reflected either numerical, verbal, or figural content, we allowed the residual variances of items sharing the same content to correlate, as it is plausible according to the BIS-model that these items share unique variance. A test scale of an ability was considered as configural invariant if the model demonstrated adequate fit in all proctoring conditions. Specifically, we interpreted a CFI > 0.95, an RMSEA < 0.06, an SRMR < 0.08, and a non-significant χ^2^-test as indicative of good model fit ([19]). Furthermore, CFI > 0.90, RMSEA < 0.08, and SRMR < 0.10 were considered as a minimally acceptable fit. Note that in some instances, it was necessary to modify the model by allowing additional residual correlations or by removing residual correlations between items sharing the same content, due to issues with non-convergence or unacceptable fit indices across all conditions. In such cases, we explicitly reported the modifications and modeled them across all conditions.

To test metric measurement invariance, we first examined a baseline model assuming configural invariance by specifying the previously established CFA models as a multigroup model across the proctoring conditions. Metric measurement invariance was then evaluated by constraining the factor loadings to be equal across conditions and comparing the fit indices of this model to the baseline model. Following [6] ([6]), changes in CFI (ΔCFI) of 0.01 or less were interpreted as indicating negligible deterioration in model fit.

Next, scalar measurement invariance was tested by constraining the intercepts of the indicators to be equal across conditions. Additionally, the mean of the latent factor for respective ability was fixed to zero in the proctored condition and estimated freely in the other conditions. The fit of the scalar model was compared to the metric model using the same fit indices and criteria. If scalar invariance was established, it was possible to compare the means of the latent factors across conditions. In this case, significant deviations from zero for the means in the UP random or UP chosen conditions would indicate that the latent means differ significantly from those in the proctored condition. If full measurement invariance could not be established, we assessed partial measurement invariance. Following the guidelines of [11] ([11]), we allowed up to 20% of the parameters to vary across groups, as this threshold is considered acceptable for establishing partial invariance.

##### Test Performance Mean Differences

To test mean differences in test performance across proctoring conditions and ability tests, we conducted a repeated measures MANOVA with proctoring condition (proctored, UP random, UP chosen) as the between-subjects factor and ability test (reasoning, short-term memory, processing speed, divergent thinking) as the within-subjects factor. The main effects of the condition and ability test as well as their interaction were tested. Tukey’s HSD tests were conducted as post hoc comparisons to further examine differences in factor scores between conditions for each ability test.

##### Test Duration

The test duration was analyzed using one-way ANOVAs with proctoring condition as the predictor. As the test was administered in two parts with a short break in between, which could differ in length between participants, we conducted the analysis separately for each part.

##### Frequency of Cheating

Proctoring records were descriptively analyzed to determine how many participants in the proctored condition could be identified as cheaters. In addition, the specific types of cheating employed by these participants were documented. Other relevant events that could have interfered with the testing situation were also descriptively analyzed.

##### Robustness Analyses

All main analyses were repeated after excluding participants from the proctored condition who were identified as cheaters based on video review as a robustness check.

## 3. Results

### 3.1. Preliminary Results

Frequency distributions for gender, current employment or student status, highest education, parental education, and first language by condition are reported in Table 4. χ^2^-tests indicated no significant differences between proctoring conditions on gender, parents’ education, and primary language (*p* > 0.05). However, χ^2^-tests indicated significant differences in employment status (χ^2^ = 18.477, *df* = 10, *p* = 0.047) and highest education (χ^2^ = 19.472, *df* = 10, *p* = 0.035). Participants in the UP random condition were less frequently full-time employed (28%) and more frequently university students (40%) compared to the Proctored condition (full-time: 47%, university: 31%) and UP chosen condition (full-time: 42%, university: 29%). Among proctored participants, the highest educational degree was more often a university degree (46%) and less often a high school diploma (Abitur; 12%) or intermediate school degree (11%), relative to the UP random (university degree: 21%, high school diploma: 20%, intermediate degree: 15%) and UP chosen conditions (university degree: 32%, high school diploma: 19%, intermediate degree: 19%). Moreover, UP random condition participants more often had a lower secondary school degree as their highest education (17%) than participants in the other two conditions (Proctored condition: 8%, UP chosen condition: 9%). However, ANOVA results (*F* = 0.152, *df* = 2, *p* = 0.859) showed that the proctoring conditions did not differ with respect to age (Proctored condition: *M* = 29.34, *SD* = 5.84; UP random condition: *M* = 29.56, *SD* = 6.51; UP chosen condition: *M* = 29.91, *SD* = 6.89).

### 3.2. Main Results

#### 3.2.1. Measurement Invariance

Model fit indices of all CFAs and measurement invariance tests are reported in Table 5. For reasoning, a one-factor model indicated a good fit for the proctored and UP random conditions and an acceptable model fit for the UP chosen condition after some modifications (for more detail, see Table 5), indicating configural measurement invariance. Metric and scalar measurement invariances could also be established.

For short-term memory, a one-factor model indicated a good fit for the proctored and UP chosen conditions and an acceptable model fit for the UP random condition after some modifications (for more detail, see Table 5), indicating configural measurement invariance. Metric measurement invariance was narrowly achieved (ΔCFI = 0.010), whereas scalar measurement invariance was not achieved (ΔCFI = 0.043). Partial scalar measurement invariance could only be achieved when 6 of 27 intercepts were freely estimated, which exceeds the 20% threshold deemed acceptable by [11] ([11]).

In detail, the following modifications had to be performed to achieve partial scalar invariance. For M_f3 (i.e., the third memory indicator referring to figural content), the proctored group intercept was freely estimated, whereas the UP random and UP chosen group intercepts were constrained to be equal. The proctored group had an intercept of 0.197 (*p* = 0.065), the UP random group had an intercept of −0.184 (*p* = 0.156), and the UP chosen group had an intercept of −0.014 (*p* = 0.893). For M_n1 (i.e., the first memory indicator referring to numerical content), the UP chosen group intercept was freely estimated, whereas the proctored and UP random group intercepts were constrained to be equal. The proctored group had an intercept of −0.333 (*p* = 0.001), the UP random group had an intercept of 0.035 (*p* = 0.773), and the UP chosen group had an intercept of 0.281 (*p* = 0.012). For M_v1 (i.e., the first memory indicator referring to verbal content), the UP chosen group intercept was freely estimated, whereas the proctored and UP random group intercepts were constrained to be equal. The proctored group had an intercept of −0.110 (*p* = 0.213), the UP random group had an intercept of −0.153 (*p* = 0.218), and the UP chosen group had an intercept of 0.254 (*p* = 0.037). For M_v2 (i.e., the second memory indicator referring to verbal content), the proctored group intercept was freely estimated, whereas the UP random and UP chosen group intercepts were constrained to be equal. The proctored group had an intercept of −0.286 (*p* = 0.003), the UP random group had an intercept of 0.094 (*p* = 0.456), and the UP chosen group had an intercept of 0.175 (*p* = 0.127). To sum up, across the four relevant test indicators, two (M_n1 and M_v2) showed lower intercepts for the proctored group compared to both UP groups, one (M_f3) showed lower intercepts for both UP groups compared to the proctored group, and one (M_v1) showed a mixed pattern, with the UP random group scoring slightly lower and the UP chosen group scoring higher than the proctored group.

For processing speed, we observed configural, metric, and scalar measurement invariance without needing any modifications.

Finally, in divergent thinking, configural measurement invariance could be demonstrated after some modifications (for more detail, see Table 5) through very good model fits across proctoring conditions. Metric and scalar measurement invariance could also be established (ΔCFI < 0.01).

Latent factor scores for reasoning, processing speed, and divergent thinking were saved for further analyses as manifest values from multigroup scalar measurement invariance models across conditions. The factor scores for short-term memory were obtained from a multigroup partial scalar invariant model and were also saved for further analyses.

#### 3.2.2. Test Performance Mean Differences

Mean differences in factor scores across conditions are depicted in Figure 3, and the exact means and standard deviations are provided in Table 6. There was no significant main effect of condition on factor scores (*F*(2, 223) = 1.29, *p* = 0.276). The within-subject factor ability test showed a significant effect (*F*(3, 221) = 8.96, *p* < 0.001). Because the factor mean for proctored participants was fixed at zero and the factor scores for the UP random and UP chosen conditions were freely estimated, this result indicates that the groups deviated from the reference category (proctored) in different ways across the various ability tests. Furthermore, the interaction between condition and ability test was significant (*F*(6, 444) = 3.06, *p* = 0.006). This indicates that the effect of proctoring condition on test performance differed by ability test. As can be seen in Figure 3, proctored participants outperformed those in the UP random and UP chosen conditions in reasoning and processing speed, whereas they performed worse in memory. For divergent thinking, participants in the UP chosen condition showed lower test scores compared to those in the proctored or UP random conditions. However, although the interaction effect was significant, Tukey post hoc tests revealed no significant differences between factor scores across conditions for any ability except for divergent thinking, where proctored participants scored significantly higher than those in the UP chosen condition (*p* = 0.046).

As an additional exploratory analysis, we repeated the same ANOVA with condition as the between-subjects factor and ability test as the within-subjects factor, this time including employment status and highest education, as well as all possible interactions, as additional factors. This was performed to test whether the results remained robust when accounting for a priori group differences reported in our preliminary analyses. In this ANOVA, employment status was not a significant predictor (*F*(5, 203) = 2.24, *p* = 0.052), while highest education was significant (*F*(4, 203) = 4.69, *p* = 0.001). However, the overall results pattern for condition and test type remained unchanged. There was no significant main effect of condition (*F*(2, 223) = 1.15, *p* = 0.318), a significant main effect of the within-subject factor ability test (*F*(3, 221) = 10.22, *p* < 0.001), and a significant interaction between condition and ability test (*F*(6, 444) = 3.27, *p* = 0.004).

#### 3.2.3. Test Duration

With respect to test duration, ANOVAs revealed no significant differences in mean duration across conditions in both the first (*F* = 1.461, *df* = 2, *p* = 0.234) and the second (*F* = 2.386, *df* = 2, *p* = 0.095) parts of the test (proctored condition: M_1_/M_2_ = 88.52/37.66, SD_1_/SD_2_ = 15.46/5.46; UP random: M_1_/M_2_ = 84.43/35.20, SD_1_/SD_2_ = 9.90/6.85; UP chosen: M_1_/M_2_ = 87.75/39.72, SD_1_/SD_2_ = 19.49/19.12).

#### 3.2.4. Frequency of Cheating

Of the 74 participants in the proctored condition, video recordings were available for all participants. Among these 74 participants, we identified three clear cases of cheating (4%). The detected cheating behaviors included taking cell phone photos of the screen, using a cell phone to solve tasks, writing down notes as a cheat sheet, and reading solutions aloud from the cell phone. In all three cases, the cheating behavior was clearly identifiable. For example, one participant took photos of the screen with a mobile phone in a mostly concealed manner, visible only when the phone was briefly raised above desk level.

Other relevant events, such as leaving the testing area, the presence of non-participating individuals, or background noises (e.g., music), occurred several times. However, such events were only classified as a major source of interference if they stood out from other participants in terms of quality or quantity (e.g., leaving the room for about ten minutes, or a combination of street noise, background music, gardening, and short conversations). Major technical issues, such as missing both video and audio output, were also categorized as relevant events with significant impact. In total, we identified 11 participants (15%) with noticeable sources of interference compared to the rest of the sample.

#### 3.2.5. Robustness Analyses

Replicating the complete main analyses after excluding the data of the three cheating participants from the proctored condition revealed no changes in the established level of measurement invariance or in latent factor score differences between the proctoring conditions, with one negligible exception. For the short-term memory test, only partial metric invariance was achieved, as one factor loading had to be freely estimated across proctoring conditions, whereas full metric invariance was established in the full sample.

## 4. Discussion

In the present study, we investigated the effects of proctoring with webcam video recordings of intelligence tests, focusing on four intelligence abilities: reasoning, short-term memory, processing speed, and divergent thinking. First, we examined measurement invariance across three proctoring conditions (proctored, UP random, and UP chosen) for each ability test separately. CFAs revealed scalar measurement invariance for reasoning, processing speed, and divergent thinking, indicating that mean-levels of these abilities are comparable across conditions. Short-term memory, however, did not show scalar measurement invariance due to substantial intercept differences across conditions. This lack of scalar invariance implies that the memory tasks did not measure the same latent construct equivalently across groups, limiting comparability.

Second, we investigated whether mean test performance differed across proctoring conditions and whether this effect varied by ability test. There was no significant main effect of proctoring condition, indicating that proctoring did not systematically increase or decrease test performance across the different abilities. However, we found a significant interaction between condition and ability test, partly in accordance with our hypotheses. Descriptively, proctored participants performed somewhat worse in short-term memory but slightly better in reasoning and processing speed compared to participants in both unproctored conditions. For divergent thinking, only participants in the UP chosen condition showed lower performance relative to the proctored condition. Nevertheless, observed differences within each ability test were of small effect size and only the difference in divergent thinking between the proctored and UP chosen conditions reached statistical significance in post hoc comparisons. Thus, despite some indication that proctoring may influence test performance, the overall effects were limited and varied by assessed ability.

Third, we investigated differences in test duration across conditions. Contrary to assumptions about cheating affecting test-taking time, no significant differences in test duration were found.

Fourth, reviewing video recordings revealed a low prevalence of observable cheating behaviors (4%) under proctored conditions, aligning with prior findings suggesting that explicit cheating remains relatively uncommon even in remote assessments. The types of observed cheating behaviors, such as writing notes or photographing the screen, are strategies typically associated with memory-based tasks, where scalar measurement invariance was not established.

### 4.1. Strengths and Limitations

Before discussing the implications of our findings for proctoring research and remote testing practice, we acknowledge strengths and limitations of the present study. A notable strength is the diverse assessment of abilities, including reasoning, short-term memory, processing speed, and divergent thinking. However, comprehension-knowledge, another important cognitive domain, was not assessed. Investigating comprehension-knowledge in future research is particularly relevant, as the cheatability of typical comprehension-knowledge tests might be especially high in unproctored remote settings due to internet access. An additional strength of the present study is the explicit assessment of measurement invariance across proctoring conditions. Indeed, our analyses demonstrated the importance of measurement invariance, as scalar invariance could not be established for memory. Therefore, in our analyses of mean differences across conditions and ability tests, we used factor scores from a partial scalar invariance model for memory, in which some intercepts were allowed to vary between conditions. It should be noted as a limitation that we slightly exceeded the commonly recommended threshold for partial measurement invariance as proposed by Dimitrov, which suggests freeing no more than 20% of the intercepts. In our model, 6 out of 27 intercepts (i.e., 22%) had to be freely estimated to achieve partial measurement invariance. Similarly to [52] ([52]), the pattern of group differences in item intercepts that were not invariant was not consistent across the indicators. While two items showed lower intercepts for the proctored group and one item showed lower intercepts for both unproctored groups, another item displayed a mixed pattern, with the UP random group showing lower and the UP chosen group showing a higher intercept than the proctored group. Moreover, it should be noted that cheating in unproctored settings is likely limited to a relatively small subsample of participants. While our measurement invariance approach captures group-level differences in test functioning across conditions, it may be less sensitive to effects that occur only in a small number of individuals. Future research could complement group-level measurement invariance approach with person-centered analyses, which may be more sensitive in detecting such subgroup effects.

Another strength of our study is the diverse sample, comprising participants aged 20 to 44 with variability in professional status and educational attainment. The experimental assignment between the proctored condition and the UP random condition ensured random allocation between these groups. However, we nevertheless observed differences between conditions in employment status and highest education. These differences may provide alternative explanations for our findings and we therefore conducted additional ANOVAs including employment status and highest education as factors. These analyses did not alter the observed effects: condition remained a non-significant between-person factor, while a significant interaction between condition and ability test persisted.

The relatively small sample size limited statistical power for detecting small mean-level differences, which possibly also explains why post hoc tests did not reveal significant differences between the conditions in any individual test despite the presence of a significant interaction of condition and test type (with the exception of divergent thinking). Additionally, the small sample size prevented the estimation of more complex CFA models incorporating all abilities simultaneously. With a larger sample, it would have been possible to investigate structural invariance among abilities, such as potential differences in the relationships between reasoning and memory across proctoring conditions. Our sample size may also have contributed to convergence issues in some models, necessitating modifications such as removing or allowing correlations between residual variances in our analyses.

Finally, the intelligence testing in this study was conducted under low stake conditions, meaning participants were neither substantially rewarded nor penalized based on their performance. Thus, participants had little extrinsic incentive to engage in cheating. Furthermore, in the present study, participants received a fixed payment regardless of performance, which may have encouraged some to complete tasks quickly and “well enough” rather than with maximal accuracy, potentially reducing the motivation to cheat. To mitigate this risk, we informed participants that their data would be checked for signs of serious engagement before payment approval. Of note, one previous study also reported cheating in low-stakes conditions, suggesting that cheating is not limited to high-stakes testing ([23], see Table 2). Although this limits generalizability to contexts involving high stake conditions (e.g., employment or university applications), our study realistically reflects common research scenarios typically characterized by low stake conditions, thus ensuring ecological validity within this context.

### 4.2. Implications and Future Research

One important finding of this study is that we observed scalar measurement invariance for all assessed ability tests except for short-term memory, suggesting that memory tasks functioned differently across proctoring conditions. Our review of recorded videos indicated that cheating behaviors primarily involved participants photographing their screens or noting down answers, which are particularly beneficial for memory tasks. If such cheating occurred in the two unproctored conditions, this could explain why scalar measurement invariance was not established for the memory test. The specific nature of the observed cheating behavior is noteworthy, as previous studies have attributed task-specific differences across proctoring conditions mainly to the ease of searchability (e.g., [23]), which facilitates cheating in unproctored settings. In contrast, our findings suggest that it was primarily the opportunity for note-taking, rather than searchability alone, that impacted the measurement properties and comparability of memory tasks across different proctoring conditions. Consequently, our findings suggest that memory tasks should not be administered without proctoring, as cheating may be particularly easy and influential in these tasks. It is also important to be careful with other tasks where taking notes could help, such as mental arithmetic problems.

In addition to searchability and note-taking, future research should also address the potential role of AI solvability in intelligence tests. Our current study found no evidence of AI-related cheating. Such cheating might have been suspected if the measurement invariance tests had revealed differences in verbal reasoning or divergent thinking tasks across proctoring conditions. However, ongoing advancements in AI and its integration into everyday applications, such as browser-based AI support, could facilitate cheating in the future. For example, advances in AI tools could mean that not only knowledge-based or vocabulary items but in some cases also more complex reasoning or problem-solving tasks might become solvable through AI assistance.

The findings regarding mean performance differences across proctoring conditions and abilities have several important implications. The absence of a significant main effect of proctoring suggests that remote testing without proctoring might generally provide comparable results to proctored testing, at least in low-stakes research settings. This is consistent with previous research on intelligence testing ([20]; [26]; [31]; [41]; [47]), which has primarily focused on reasoning tests. Thus, both our study and most prior research on proctoring effects in intelligence testing contradict the findings of [44]’s ([44]) meta-analysis, which was based primarily on studies conducted in educational contexts and which reported that unproctored participants outperformed those in proctored conditions. This suggests that such effects may not generalize to intelligence testing.

However, the significant interaction between condition and ability test highlights that proctoring effects may vary by ability, and importantly, differences may not always disadvantage proctored participants. In our analysis, we observed descriptive advantages for proctored participants in reasoning and processing speed, as well as a significant advantage in divergent thinking compared to the UP chosen group. Similar performance benefits under proctored conditions have previously been reported (e.g., [44]; [1]) and could be explained by motivational factors or increased concentration in supervised settings. As the observed differences across proctoring conditions varied across ability tests, it is possible that the motivational impact of webcam monitoring differs across ability domains, with some tasks that rely more heavily on effort or concentration benefiting more than others. The idea is also in line with some previous findings of superior performance in proctored conditions for tests of low cheatability ([8]; [51]; see Table 2).

This suggests that researchers should exercise caution when interpreting the results of unproctored tests, even for tasks that are difficult to cheat at, as participants may not exert maximum effort or concentration in unsupervised conditions.

Furthermore, we found no differences in test duration between the proctoring conditions, which argues against the occurrence of extensive cheating in the unproctored condition, as such behavior would likely have resulted in longer test durations. One possible explanation for these findings is that cheating in most of the tasks used in our study (i.e., reasoning, processing speed, and divergent thinking) is unlikely, as the solutions are not easily searchable or amenable to note-taking. Alternatively, the finding of no differences in test duration between the proctoring conditions could be explained by a very low amount of cheating overall. As expected, within the proctored condition, we identified only a very small proportion of cheaters (4%), which is in line with previous research on intelligence tests that also found little or no evidence of cheating ([26]; [31]; [52]). Note that in the present study, webcam-based proctoring involved only video recording of the participant without screen capture. While this approach can effectively detect overt behaviors such as note-taking or photographing the screen, it is less capable of identifying covert behaviors, including online searches or the use of AI-assisted tools. Other forms of proctoring such as live remote monitoring, screen recording, or in-person supervision may offer broader coverage of potential cheating behaviors, but they also come with higher costs, greater intrusiveness, and possible privacy concerns.

To conclude, cheating appears to happen rarely in low-stakes settings when using complex intelligence tests, and we are therefore generally optimistic about the validity of intelligence testing in online remote environments. However, when memory tasks are included, we strongly recommend the use of proctoring, as cheating cannot be ruled out due to the ease of note-taking. The same caution applies to tasks that are highly searchable ([44]), such as knowledge questions in comprehension-knowledge tests. Moreover, advantages for proctored participants, potentially due to increased motivation or higher concentration, should also be considered when choosing the testing environment. Therefore, if proctoring is feasible, we generally recommend its implementation. For complex tasks, such as reasoning or divergent thinking, or tasks where cheating is inherently difficult, such as processing speed, we remain optimistic that cheating is rare and negligible even without proctoring. Finally, when comparing proctored and unproctored conditions, we strongly recommend conducting measurement invariance tests before comparing test means across groups. This is crucial, as tests may differ not only in their mean scores across proctoring conditions, but also in whether the same indicators measure the underlying ability construct in a comparable way. Overall, researchers and practitioners should carefully consider test-specific vulnerabilities when choosing between proctored and unproctored online cognitive assessments.

## Figures and Tables

**Figure 1 jintelligence-13-00110-f001:**
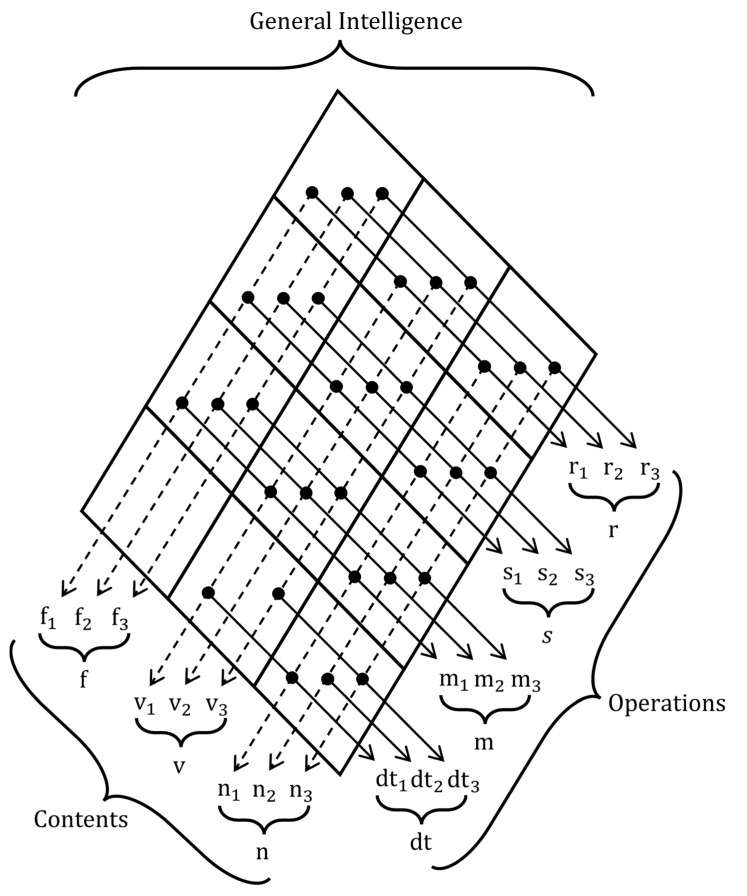
BIS+ indicators based on the berlin intelligence structure model. Note. r = reasoning, s = processing speed, m = short-term memory, dt = divergent thinking, f = figural ability, v = verbal ability, n = numerical ability.

**Figure 2 jintelligence-13-00110-f002:**
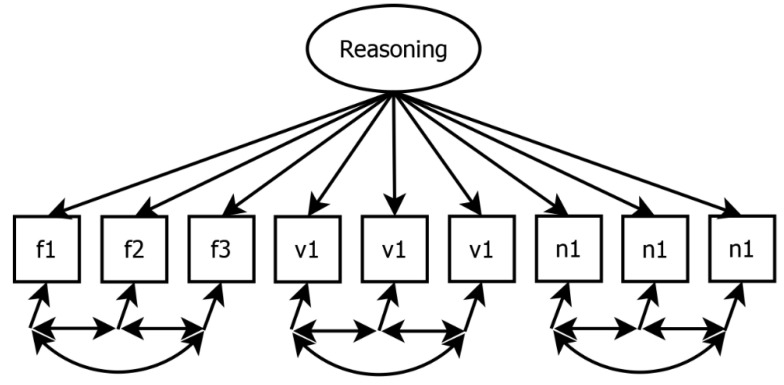
Confirmatory factor analysis of resoning based on nine content specific indicators. Note. f = figural ability, v = verbal ability, n = numerical ability. Each indicator represents reasoning ability and refers to either figural, verbal, or numerical content. Residual correlations between indicators with the same content domain were admitted.

**Figure 3 jintelligence-13-00110-f003:**
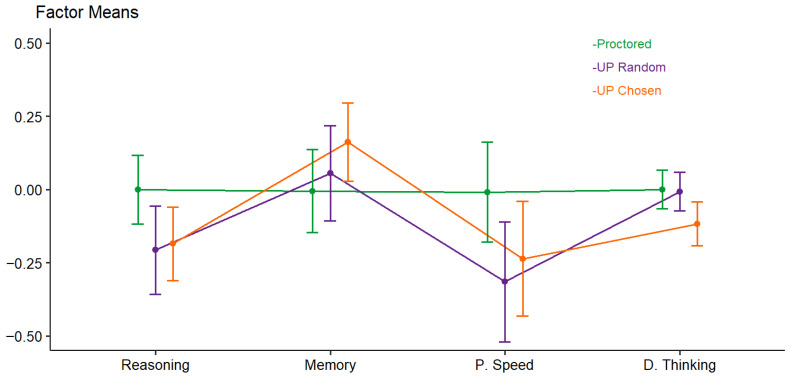
Mean differences in intelligence factor scores by proctoring condition. Note. P. Speed = processing speed. D. Thinking = divergent thinking. All factor mean scores except for short-term memory stemmed from scalar measurement invariance multigroup confirmatory factor analyses with proctoring condition as the multigroup factor. Short-term memory factor scores stemmed from the partial scalar measurement invariant multigroup confirmatory factor analysis.

**Table 1 jintelligence-13-00110-t001:** Overview of recent studies testing the effects of proctoring.

Authors (Year)	Assessment Type or Paradigm	Proctoring Solution	Design	*N*	Mean Age	Age Range	Population	Country	Stakes	Average Performance	Cheating	Test Duration	Measurement Invariance
[1] ([1])	academic testing	software	within	97			university students	US	high	unproctored > proctored	not tested	unproctored > proctored	not tested
[2] ([2])	academic testing	software	within	101		19–21	university students	Indonesia	high	unproctored > proctored	not tested	not tested	not tested
[4] ([4])	academic testing	on-site proctoring	within	2010			university students	US	high	no difference	yes (ineffective at boosting performance)	not tested	not tested
[5] ([5])	academic testing	on-site proctoring	within	510			university students	US	high	unproctored > proctored	yes (increasing over the semester)	not tested	not tested
[9] ([9])	academic testing	video monitoring	within	1694		mostly 30–50	university students (with employment)	US	high	unproctored > proctored	not tested	unproctored > proctored	not tested
[10] ([10])	academic testing	video recording	between, not random	648	19.99–21.40		university students	US	high	unproctored > proctored	yes (prior to proctoring)	not tested	not tested
[12] ([12])	academic testing	on-site proctoring	between, not random	1584	23.52	18–39	general population	Spain	low	unproctored > proctored	not tested	no difference	not tested
[15] ([15])	academic testing	on-site proctoring	within	850	26		university students	US	high	Set 1: no difference Set 2: proctored > unproctored	not tested	not tested	not tested
[31] ([31])	cognitive ability testing	on-site proctoring	within	487	21	19–27	Norwegian Armed Forces	Norway	high	no difference	yes (minimal)	not tested	Strict
[32] ([32])	academic testing	software	between, not random	426			university students	US	high	unproctored > proctored	not tested	not tested	not tested
[33] ([33])	academic testing	software	between, not random	252			university students	US	high	mixed	not tested	not tested	not tested
[37] ([37])	academic testing	software; on-site proctoring	between, not random	72			university students	US	high	unproctored > proctored	not tested	not tested	not tested
[39] ([39])	academic testing	software; on-site proctoring	between, not random	296		25–63	university students (postgraduates)	Mexico	high	unproctored > proctored	not tested	not tested	not tested
[53] ([53])	academic testing	on-site proctoring	between, not random	234			university students	US	Sample 1: high, Sample 2: low	Sample 1: unproctored > proctored Sample 2: proctored > unproctored	not tested	unproctored > proctored	not tested
[54] ([54])	fake question paradigm	on-site proctoring	between, randomized	177/158/191/166		19–22	university students	China	high	not tested	proctored > unproctored	not tested	not tested

**Table 2 jintelligence-13-00110-t002:** Overview of studies investigating the effects of proctoring in intelligence test settings.

Authors (Year)	Cognitive Tests	Proctoring Solution	Design	*N*	Mean Age	Age Range	Population	Country	Stakes	Average Performance	Cheating	Test Duration	Measurement Invariance
[8] ([8])	numerical, verbal, and figural reasoning	on-site proctoring	within	86/ 64	31.51/ 22.77		professionals/university students	UK	low	proctored > unproctored	not tested	not tested	not tested
[20] ([20])	figural matrices	on-site proctoring	between, not random	481	21.7/ 24.7/ 27.0	18–46/ 18–50/ 18–56	mixed	Germany	low	no difference	not tested	not tested	not tested
[23] ([23])	Graduate Record Examination test/ Raven’s matrices	video recording	between, randomized	295	30		general population	US	low	searchable: unproctored > proctored not searchable: no difference	unproctored > proctored	not tested	not tested
[26] ([26])	numerical and verbal reasoning	on-site proctoring	within	ca. 3500		Mostly 21–29	applicants	UK	high	no difference	low frequency of aberrant scores (0.3–2.2%)	not tested	not tested
[31] ([31])	figure matrices, word similarities, number reasoning/mental rotation	on-site proctoring	within	487	21	19–27	Norwegian Armed Forces	Norway	high	no difference	yes (minimal)		strict
[41] ([41])	figure series, matrices, number series	on-site proctoring	within	425	27		applicants	Netherlands	high	no difference	medium frequency of aberrant scores (5–12%)		not tested
[47] ([47])	creativity	on-site proctoring	within and between (randomized)	163	21.9	19–25	potential applicants	Singapore	low	no difference			
[51] ([51])	spatial ability, reasoning	on-site proctoring	between, randomized	457	19.64	18–32	university students	USA	low	spatial: mixed reasoning: proctored > unproctored	not tested	not tested	not tested
[52] ([52])	deductive reasoning	software	between, not randomized	24,750/ 9615			applicants	USA	high	unproctored > proctored/ proctored > unproctored	no evidence of cheating	proctored > unproctored/ unproctored > proctored	differential item functioning in 8 of 30 indicators

**Table 3 jintelligence-13-00110-t003:** Description of the assessed intelligence ability scales.

Ability	Description	Example Task
**Reasoning**	To understand new information, identify connections, and draw conclusions through thoughtful analysis.	Solving figural analogies.
**Short-Term Memory**	To quickly memorize information and then recall or recognize it correctly.	Memorize two-digit numbers and freely recall them.
**Processing Speed**	To stay focused and work quickly and accurately on simple tasks that require little deliberation.	Deciding whether a word (e.g., oak) is part of a category (e.g., tree) or not.
**Divergent Thinking**	To generate many original ideas by approaching problems from different perspectives.	Naming abilities a salesman should not have.

**Table 4 jintelligence-13-00110-t004:** Frequency Distribution by Condition.

Variable	Category	Proctored	Un-Proctored	No Cam	χ^2^	*df*	*p*
Gender	Female	19/26%	23/31%	33/43%	5.707	4	0.222
	Male	53/72%	51/68%	43/56%			
	Non-binary	2/3%	1/1%	1/1%			
Employment Status	Full-time employed	35/47%	21/28%	32/42%	18.477	10	0.047
	Part-time employed	9/12%	6/8%	8/10%			
	Self-employed	1/1%	7/9%	4/5%			
	Unemployed	6/8%	4/5%	3/4%			
	University student	23/31%	30/40%	22/29%			
	Others	0/0%	7/9%	7/9%			
Highest Education	High school diploma (Abitur)	9/12%	15/20%	15/19%	19.472	10	0.035
	Intermediate school degree	8/11%	11/15%	15/19%			
	Lower secondary school	6/8%	13/17%	7/9%			
	No school degree	15/20%	13/17%	13/17%			
	University degree	34/46%	16/21%	25/32%			
	Others	2/3%	5/7%	0/0%			
Parents Education	Doctorate (PhD)	4/5%	5/7%	3/4%	11.679	12	0.307
	High school diploma (Abitur)	17/23%	12/16%	12/16%			
	Intermediate school degree	16/22%	18/24%	22/29%			
	Lower secondary school	7/9%	7/9%	16/21%			
	Others	2/3%	2/3%	4/5%			
	University degree	28/38%	30/40%	19/25%			
Primal Language	German	63/85%	64/85%	64/83%	0.175	2	0.916
	Others	11/15%	11/15%	13/17%			

**Table 5 jintelligence-13-00110-t005:** Fit indices from confirmatory factor analyses and measurement invariance testing.

Model	*n*	χ^2^	SF	*df*	*p*	CFI	RMSEA	SRMR	Δχ^2^	Δ*df*	Δ*p*	ΔCFI	Δ RMSEA	Δ SRMR	AIC	BIC
**Reasoning Ability**																
Configural MI																
Proctored ^1^	74	22.843	0.969	19	0.244	0.979	0.050	0.044							1671	1752
UP random ^1^	75	19.302	0.945	19	0.438	1	0	0.041							1799	1880
UP chosen ^1^	77	31.675	0.956	19	0.034	0.925	0.096	0.061							1810	1892
Baseline	226	73.86	0.957	57	0.066	0.97	0.060	0.049							5281	5640
Metric MI	226	91.559	0.989	73	0.070	0.962	0.060	0.087	18.013	16	0.323	−0.008	0	0.038	5269	5573
Scalar MI	226	102.275	0.992	89	0.159	0.973	0.046	0.092	10.843	16	0.819	0.011	−0.014	0.005	5247	5497
**Memory Ability**																
Configural MI																
Proctored ^2^	74	17.75	0.977	15	0.276	0.988	0.044	0.043							1564	1654
UP random ^2^	75	22.588	1.033	15	0.093	0.961	0.08	0.055							1858	1949
UP chosen ^2^	77	6.984	1.031	15	0.958	1	0	0.034	0	0	1	0.039	−0.08	−0.021	1794	1885
Baseline	226	47.234	1.014	45	0.381	0.999	0.015	0.044	40.472	30	0.096	−0.001	0.015	0.010	5217	5617
Metric MI	226	67.831	1.013	61	0.256	0.989	0.037	0.081	20.608	16	0.194	−0.010	0.022	0.037	5206	5551
Scalar MI	226	106.412	1.005	77	0.015	0.946	0.071	0.099	39.232	16	0.001	−0.043	0.034	0.018	5212	5503
P. Scalar MI ^3^	226	81.134	1.016	70	0.171	0.980	0.045	0.083	13.238	9	0.152	−0.009	0.008	0.002	5201	5516
**Processing Speed Ability**																
Configural MI																
Proctored	74	23.875	0.927	18	0.159	0.981	0.069	0.040							1444	1527
UP random	75	25.807	0.987	18	0.104	0.968	0.093	0.047							1596	1679
UP chosen	77	20.643	0.901	18	0.298	0.993	0.042	0.036	0	0	1	0.025	−0.051	−0.011	1637	1722
Baseline	226	70.554	0.938	54	0.065	0.981	0.071	0.041	49.744	36	0.063	−0.012	0.029	0.005	4678	5047
Metric MI	226	96.101	0.924	70	0.021	0.971	0.075	0.087	25.797	16	0.057	−0.01	0.004	0.046	4668	4983
Scalar MI	226	109.272	0.941	86	0.046	0.974	0.065	0.090	13.815	16	0.612	0.003	−0.01	0.003	4651	4910
**Divergent Thinking Ability**																
Configural MI																
Proctored ^4^	74	0.397	0.98	1	0.529	1	0	0.014							912	956
UP random ^4^	75	1.163	0.793	1	0.281	1	0	0.022							993	1037
UP chosen ^4^	77	2.358	0.89	1	0.125	0.978	0.12	0.029	0	0	1	−0.022	0.12	0.007	1018	1062
Baseline	226	3.842	0.888	3	0.279	1	0	0.022	1.48	2	0.477	0.022	−0.12	−0.007	2924	3119
Metric MI	226	10.343	1.011	11	0.500	1	0	0.049	6.664	8	0.573	0	0	0.027	2915	3082
Scalar MI	226	15.241	0.987	19	0.707	1	0	0.058	4.807	8	0.778	0	0	0.009	2903	3043

Note. MI = measurement invariance. UP = unproctored. ^1^ = Models were modified by removing one residual correlation between two indicators referring to figural content. ^2^ = Models were modified by allowing three additional residual correlations suggested by modification indices. ^3^ = Partial scalar MI was achieved after allowing six of 27 intercepts to vary between proctoring conditions. ^4^ = Models were modified by allowing three additional residual correlations suggested by modification indices.

**Table 6 jintelligence-13-00110-t006:** Latent correlations among intelligence abilities and their mean scores across proctoring conditions.

Variable	Proctored		UP Random		UP Chosen		1	2	3
	*M*	*SD*	*M*	*SD*	*M*	*SD*			
1. Reasoning	0.00	0.51	−0.21	0.66	−0.19	0.55			
2. Memory	−0.01	0.61	0.06	0.71	0.16	0.59	0.41 **		
							[0.29, 0.51]		
3. Processing Speed	−0.01	0.74	−0.32	0.89	−0.24	0.86	0.71 **	0.40 **	
							[0.64, 0.77]	[0.28, 0.50]	
4. Divergent Thinking	0.00	0.28	−0.01	0.29	−0.12	0.33	0.35 **	0.20 **	0.44 **
							[0.23, 0.46]	[0.08, 0.33]	[0.33, 0.54]

Note. ** *p* < 0.01. Values in square brackets indicate the 95% confidence interval for each correlation. Latent factor scores for reasoning, processing speed, and divergent thinking were obtained from multigroup scalar measurement invariant models across conditions. The factor scores for memory were obtained from a multigroup partial scalar invariance model. In these models, the respective factor score for the proctored condition was fixed at 0, whereas the factor scores for the UP random and UP chosen conditions were estimated freely.

## Data Availability

Analysis code and dataset are available online on: https://osf.io/yznct/?view_only=66c66d3627104459926aaf4afa265c74 (Accessed on 26 August 2025).

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
