# Peer review of "Effects of Proctoring on Online Intelligence Measurement: A Literature Overview and an Empirical Study"

_jintelligence, 2025, doi:10.3390/jintelligence13090110_

Round 1
Reviewer 1 Report
Comments and Suggestions for Authors
This is a nice study that provides insight both into methodological ways of looking at measurement implications of proctoring while also lending some insight into practical consequences. I provide some considerations below for possibly improving the paper:
1. It is unclear to me how well the experiment represents what readers would
view as online proctoring, as it seems the only difference manipulated was
whether the camera was on or not. Were participants informed in any way as
to what was considered permissible and non-permissible test-taking behavior?
The observation of several "cheating" instances in the proctoring condition makes
one wonder whether this was actual cheating or different understandings of the rules. As these tests are low stakes there is naturally also no major incentive for cheating behavior. This is noted in the paper, but seems important to highlight. The study would remain important for understanding whether monitoring is important in these types of assessments, but one would worry about relevance to high stakes settings.
2. As noted also in the literature review, the potential to see better performances in the proctoring condition on certain tests would suggest the video monitoring is perhaps also motivational for certain tests (subjects display better test-taking behavior when knowing they are watched). It seems these two aspects of the video monitoring (motivational from being watched and reduced potential
for cheating) are perhaps at play to varying extents for all of the assessments?
3. It would seem useful to talk more specifically about what was seen for the items in which invariance violations were found, and whether the observations made sense (e.g., were items that might easier to cheat on found to produce violations to scalar invariance?)
4. I appreciate the ability to explore issues of proctoring using invariance analyses, although it seems that most perspectives on cheating in unproctored settings recognize the cheating not as a sample-wide phenomenon, but as one restricted to a possibly small subsample of participants that choose to cheat It would be helpful to talk about this issue further, as it seems other forms of person-centered analyses (i.e., that classify individuals across conditions as cheaters/noncheaters) might display greater statistical power and ultimately show more cheaters being detected in the unproctored conditions.
Author Response
Reviewer 1
This is a nice study that provides insight both into methodological ways of looking at measurement implications of proctoring while also lending some insight into practical consequences. I provide some considerations below for possibly improving the paper:
Thank you for this overall positive evaluation
1. It is unclear to me how well the experiment represents what readers would
view as online proctoring, as it seems the only difference manipulated was
whether the camera was on or not. Were participants informed in any way as
to what was considered permissible and non-permissible test-taking behavior?
The observation of several "cheating" instances in the proctoring condition makes
one wonder whether this was actual cheating or different understandings of the rules. As these tests are low stakes there is naturally also no major incentive for cheating behavior. This is noted in the paper, but seems important to highlight. The study would remain important for understanding whether monitoring is important in these types of assessments, but one would worry about relevance to high stakes settings.
Thank you for pointing this out. In the proctored condition, participants were explicitly informed prior to testing that the purpose of webcam monitoring was to create a standardized testing environment and “to ensure that participation takes place independently and without the use of impermissible aids”. However, the rules were not operationalized in a list of specific permitted and prohibited behaviors, in line with typical low-stakes research practice. We added this information into Section 2.3.2 (see p. 17).
The three participants who were identified as engaging in cheating behavior in the proctored condition were unambiguous cases. For example, one participant took photos of the screen with a mobile phone in a way that was mostly concealed from view, except in some trials where the phone was raised high enough to be visible on camera. We have now clarified these aspects in the description of the proctoring procedure (see p. 27) and noted that the absence of a detailed rule list might have contributed to occasional misunderstandings of permissible behavior.
We also agree that the low-stakes nature of the present study implies limited incentives for cheating. This point was already noted in the discussion, but we have now made it more explicit in the Strengths and Limitations section (see p. 30). Of note, one previous study also reported cheating in low-stakes conditions, suggesting that cheating is not limited to high-stakes testing (Karim et al., 2014, see Table 2).
2. As noted also in the literature review, the potential to see better performances in the proctoring condition on certain tests would suggest the video monitoring is perhaps also motivational for certain tests (subjects display better test-taking behavior when knowing they are watched). It seems these two aspects of the video monitoring (motivational from being watched and reduced potential for cheating) are perhaps at play to varying extents for all of the assessments?
This is an interesting idea. We have already noted in our discussion (Section 4.2) that webcam monitoring could possibly increase participant motivation or concentration and consequently improve performance in some tasks. We now explicitly acknowledge that such motivational effects could vary in magnitude depending on the specific ability tested (see p. 31). The idea is also in line with some previous findings of superior performance in proctored conditions for tests of low cheatability (Coyne et al., 2005; Williamson III et al., 2017; see Table 2).
3. It would seem useful to talk more specifically about what was seen for the items in which invariance violations were found, and whether the observations made sense (e.g., were items that might easier to cheat on found to produce violations to scalar invariance?)
We have expanded the Results section to describe in detail the specific items for which intercepts were freed to achieve partial scalar invariance (see p. 22- 23). For each affected indicator (M_f3, M_n1, M_v1, M_v2), we now report the group-specific intercept estimates and note whether the proctored group or the unproctored groups had higher or lower intercepts. As can be seen, the pattern was not consistent across items: Two items (M_n1, M_v2) showed lower intercepts for the proctored group compared to both unproctored groups, one item (M_f3) showed lower intercepts for both unproctored groups, and one item (M_v1) displayed a mixed pattern, with one unproctored group lower and the other showing higher intercepts than the proctored group. In line with Wright et al. (2014), we discuss this inconsistency in the Limitations section (p. 29) and note that it does not allow for a clear conclusion as to whether items with noninvariant intercepts are systematically easier or harder for specific groups.
4. I appreciate the ability to explore issues of proctoring using invariance analyses, although it seems that most perspectives on cheating in unproctored settings recognize the cheating not as a sample-wide phenomenon, but as one restricted to a possibly small subsample of participants that choose to cheat It would be helpful to talk about this issue further, as it seems other forms of person-centered analyses (i.e., that classify individuals across conditions as cheaters/noncheaters) might display greater statistical power and ultimately show more cheaters being detected in the unproctored conditions.
Thank you for this suggestion. We agree that cheating in unproctored conditions is likely limited to a relatively small subsample of participants, and that a variable-centered approach such as measurement invariance testing may be limited in detecting differences that apply only to a few individuals. Person-centered analyses could therefore provide further insights in the differences between conditions. We have now addressed this point in the limitation section, noting that future research might combine invariance testing with person-centered approaches to capture such effects more directly (see p. 29).
Reviewer 2 Report
Comments and Suggestions for Authors
The current study compared web-cam proctored and unproctored conditions on a large intelligence test battery. It also includes some reviews of recent work.
Impressive features include the review, use of a comprehensive test battery, and visual coding of webcam data for cheating.
Limitations which are noted include (a) modest sample size, and (b) the fact that it was conducted in a low-stakes context.
In general, the paper is clearly written and presents results transparently.
It seems like it would add value to this emerging literature.
ASSORTED POINTS
- Regarding " primarily investigated reasoning tasks". I wasn't clear what you mean by this. Do you mean "matrix reasoning"?
- I think the abstract should possibly specify more precisely the nature of the prctor: i.e., is it just webcam or does it also include screen capture, etc.
- In the abstract, clarify the context in which participants completed the assessments. Were they motivated to cheat in some way?
- Regarding " We conclude that proctoring is crucial for easily cheatable tasks, such as memory tasks, but less critical for complex cognitive tasks"; Does chatgpt change this? Previously it might be things like using calculators when you had to do things in your head, or googling knowledge and vocabulary questions. But now chatgpt is pretty good at a wide range of psychometric items. And some of thsoe might constitute complex cognitive tasks.
- Regarding " Implementing a proctoring solution in remote testing requires participants' willingness to be monitored via a webcam feed and increases costs and participation burden compared to unproctored assessments"... It's probably important to be clear that online proctoring is more than just webcams. I.e., screen capture, key press logging, etc.
- In the opening, you may want to define cheating and provide an overview of the main classes of cheating (e.g., using disallowed materials; getting another person to help or sit the test; getting prior access to test battery, etc.)
- Regarding "A key limi- tation, however, is that the original studies did not report measurement invariance of the tests across conditions." You may want to flag why that is interesting and why you would expect lack of measurement invariance (e.g., in some contexts, items within a test vary in cheatability)
- The review of recent literature is a nice inclusion. the average performance columns would probably be better as a cohen's d.
- I guess as a general reflection on Prolific testing, the incentive is often to do it quickly and perhaps do it well enough to still get paid. That said, feedback or intrinsic interest in such test might also be relevant. I think whenever you study cheating, you have to think about whether there is an incentive to cheat. You may want to note this in the method, discuss in discussion, and provide a contextual description in the abstrac.
- It's great that you provide data anlaysis code online. Is it possible to provide the data (or some of form of the data) on the OSF.
- I feel like Figure 3 is hard to read and that a simple table of means and sds would provide clearer interpretation.
- I wonder whether some discussion of webcam versus other forms of proctoring would be worth considering. You could also consider how it relates to different forms of cheating.
Author Response
Reviewer 2
The current study compared web-cam proctored and unproctored conditions on a large intelligence test battery. It also includes some reviews of recent work.
Impressive features include the review, use of a comprehensive test battery, and visual coding of webcam data for cheating.
Limitations which are noted include (a) modest sample size, and (b) the fact that it was conducted in a low-stakes context.
In general, the paper is clearly written and presents results transparently.
It seems like it would add value to this emerging literature.
Thank you for your positive feedback and your helpful suggestions!
ASSORTED POINTS
1. Regarding " primarily investigated reasoning tasks". I wasn't clear what you mean by this. Do you mean "matrix reasoning"?
Thank you for pointing out this ambiguity. We have clarified in the manuscript that by “reasoning tasks” we refer to a broader class of reasoning tests across different content domains, not only to matrix reasoning. We clarified this in the “Proctoring in Intelligence Testing” section (see p. 8).
2. I think the abstract should possibly specify more precisely the nature of the prctor: i.e., is it just webcam or does it also include screen capture, etc.
Thank you for pointing this out. We have now clarified in the abstract that the proctoring consisted solely of webcam video recording of the participants without screen capture.
3. In the abstract, clarify the context in which participants completed the assessments. Were they motivated to cheat in some way?
Thank you for this comment. We have clarified in the abstract that the assessments were conducted in a low-stakes research context, where participants had no substantial external incentives to cheat.
4. Regarding " We conclude that proctoring is crucial for easily cheatable tasks, such as memory tasks, but less critical for complex cognitive tasks"; Does chatgpt change this? Previously it might be things like using calculators when you had to do things in your head, or googling knowledge and vocabulary questions. But now chatgpt is pretty good at a wide range of psychometric items. And some of thsoe might constitute complex cognitive tasks.
We completely agree with this evaluation, and we already address this issue in our discussion (Section 4.2,) where we note the potential for AI-assisted cheating and its implications for tasks such as verbal reasoning or divergent thinking. We have now added a brief example to explicitly reference the possibility that AI could successfully solve some types of complex psychometric tasks (see p. 31).
5. Regarding " Implementing a proctoring solution in remote testing requires participants' willingness to be monitored via a webcam feed and increases costs and participation burden compared to unproctored assessments"... It's probably important to be clear that online proctoring is more than just webcams. I.e., screen capture, key press logging, etc.
Thank you for pointing this out. We have clarified that online proctoring can involve more than webcam monitoring, including methods such as screen capture or key press logging (see p. 2). In the Methods section, we also explicitly note that additional proctoring methods, such as screen capture or keystroke logging, were not implemented due to limitations in technical feasibility within the Pavlovia system and their potential to reduce participant acceptance (see p. 13).
6. In the opening, you may want to define cheating and provide an overview of the main classes of cheating (e.g., using disallowed materials; getting another person to help or sit the test; getting prior access to test battery, etc.)
Thank you for this comment. We have revised the opening section to include a short definition of cheating in the context of intelligence testing, along with concrete examples of cheating behavior (see p. 1).
7. Regarding "A key limi- tation, however, is that the original studies did not report measurement invariance of the tests across conditions." You may want to flag why that is interesting and why you would expect lack of measurement invariance (e.g., in some contexts, items within a test vary in cheatability)
Thank you for this helpful suggestion. We have now expanded our rationale for why the absence of reported measurement invariance is important and have outlined why lack of invariance might be expected in the context of proctored versus unproctored testing (see p. 3).
8. The review of recent literature is a nice inclusion. the average performance columns would probably be better as a cohen's d.
Thank you for this positive evaluation and this suggestion. We revisited the original studies, but unfortunately, in many cases the reported results (e.g., regression coefficients or ANOVA outcomes) do not provide the necessary descriptive statistics (means and standard deviations) to compute Cohen’s d. Therefore, it was not feasible to include a column with standardized mean differences across the reviewed studies.
9. I guess as a general reflection on Prolific testing, the incentive is often to do it quickly and perhaps do it well enough to still get paid. That said, feedback or intrinsic interest in such test might also be relevant. I think whenever you study cheating, you have to think about whether there is an incentive to cheat. You may want to note this in the method, discuss in discussion, and provide a contextual description in the abstrac.
We agree that incentives in online testing platforms such as Prolific can influence participants’ motivation and potential cheating behavior. We have now addressed this point in the abstract, methods, and discussion sections (see p. 13 and p. 30).
10. It's great that you provide data anlaysis code online. Is it possible to provide the data (or some of form of the data) on the OSF.
Thank you for pointing this out. We have uploaded the dataset to the OSF repository and indicated this in the Methods section.
11. I feel like Figure 3 is hard to read and that a simple table of means and sds would provide clearer interpretation.
Thank you for this comment. We have modified Figure 3 by slightly shifting the groups along the horizontal axis, and we believe it has become much clearer as a result. Furthermore, we think that the revised version provides a clear visual representation of the interaction effect at a glance, which is why we would like to retain this version. Please note that the means and standard deviations were already presented in Table 6. We have now made this more explicit in the manuscript (see p. 26).
12. I wonder whether some discussion of webcam versus other forms of proctoring would be worth considering. You could also consider how it relates to different forms of cheating.
Thank you for this suggestion. We have now added a brief discussion comparing webcam-based proctoring with other forms of proctoring (e.g., live remote monitoring, screen capture, and in-person supervision) and how these may differentially address various forms of cheating (see p. 32).
Round 2
Reviewer 1 Report
Comments and Suggestions for Authors
I appreciate the authors' responses to my comments, and believe the paper can now be published.
Reviewer 2 Report
Comments and Suggestions for Authors
The authors have addressed my concerns from the previous revision.
REVIEW OF PREVIOUS POINTS
- Okay
- Okay
- Okay
- It's a pity. It might have been possible to do your best to extract such values. But Okay.
- Excellent!